# Research on Fine-Grained Image Recognition of Birds Based on Improved YOLOv5

**DOI:** 10.3390/s23198204

**Published:** 2023-09-30

**Authors:** Xiaomei Yi, Cheng Qian, Peng Wu, Brian Tapiwanashe Maponde, Tengteng Jiang, Wenying Ge

**Affiliations:** College of Mathematics & Computer Science, Zhejiang A & F University, Hangzhou 311300, China; yxm@zafu.edu.cn (X.Y.); wonder357182466@gmail.com (C.Q.); brianyb312@gmail.com (B.T.M.); 202205100214@stu.zafu.edu.cn (T.J.); gwy8817@stu.zafu.edu.cn (W.G.)

**Keywords:** bird identification, fine-grained, part-based, YOLOv5, Res2Net-CBAM

## Abstract

Birds play a vital role in maintaining biodiversity. Accurate identification of bird species is essential for conducting biodiversity surveys. However, fine-grained image recognition of birds encounters challenges due to large within-class differences and small inter-class differences. To solve this problem, our study took a part-based approach, dividing the identification task into two parts: part detection and identification classification. We proposed an improved bird part detection algorithm based on YOLOv5, which can handle partial overlap and complex environmental conditions between part objects. The backbone network incorporates the Res2Net-CBAM module to enhance the receptive fields of each network layer, strengthen the channel characteristics, and improve the sensitivity of the model to important information. Additionally, in order to boost data on features extraction and channel self-regulation, we have integrated CBAM attention mechanisms into the neck. The success rate of our suggested model, according to experimental findings, is 86.6%, 1.2% greater than the accuracy of the original model. Furthermore, when compared with other algorithms, our model’s accuracy shows noticeable improvement. These results show how useful the method we suggested is for quickly and precisely recognizing different bird species.

## 1. Introduction

A bird diversity survey is an important indicator of ecological diversity, and birds are often used by researchers to assess the ecological environment due to their sensitivity to environmental changes and easy field investigation [1]. For example, environmental monitoring relies on changes in bird populations [2], climate change monitoring considers changes in bird community composition [3], and biodiversity surveys focus on bird populations [4]. All of the above studies require the identification and monitoring of bird species and use the identification results to determine environmental changes, such as determining whether the wetland is degraded by measuring the number of bird populations present in a certain wetland.

In recent years, advances in machine learning have led to the proposal of deep learning-based bird classification methods, including audio features for classification and recognition [5,6,7] and image features [8,9,10]. Among them, bird classification methods relying on audio features are easily disrupted by environmental noise, necessitating audio denoising as a prerequisite in practical applications. This requirement makes the identification task more challenging. In contrast, the bird classification method based on image features is less affected by environmental factors. This aspect facilitates the preliminary screening of bird images through content captured by camera equipment.

Additionally, image recognition models can automatically detect birds through mobile or field devices. The development of high-precision and lightweight bird image classification models holds significant research value for cost-effective bird detection.

Due to the characteristics of large intra-class differences and small inter-class differences in bird images, they can serve as key features for differentiation. These features are typically distributed in imperceptible locations. Early bird image recognition predominantly employs multi-layer classification algorithms based on traditional artificial features. In the CUB200-2011 bird dataset report published by Wah et al. [11], they utilized selective local localization and classification recognition based on a semantic part model, employing a substantial amount of annotated information. Berg et al. [12] subsequently proposed an algorithm capable of automatically searching for effective features in images, which can be used for distinguishing classifications. They also introduced a two-step strategy of first locating and then identifying.

Yao, Yang, et al. [13,14] attempted to reduce the algorithm’s complexity by utilizing template matching instead of sliding windows. Donahue et al. [15] applied CNNs to fine-grained image recognition research and discovered that CNNs exhibit strong generalization capabilities in this study. Following this, part-based R-CNN [16] was introduced, employing a CNN architecture from the local area detection network for feature extraction. This architecture constitutes a multi-level classification model based on CNN features. Zhang et al. [17] modeled small semantic information after labeling and introduced new semantic part information in the classification network. This enabled the positioning of multiple semantic parts and overall recognition, yielding favorable results in target identification. Wang et al. [18] constructed a model structure involving patch relationships. This structure automatically extracts internal information that is distinguishable and contains discernible semantic part information for modeling. Along with the part-based classification method discussed above, researchers also performed classified recognition based on the idea of blending the local location function and feature extraction function in recognition of images into the CNN structure. A full-fledged classification method was created as a result of this. The end-to-end model based on a recurrent neural network used a recurrent neural network (RNN) [19,20] to achieve image classification.

By using high-order picture information, the end-to-end approach based on high-order pooling [21,22] sought to create more discriminative image features to complete recognition tasks. To put it simply, getting second- or higher-order statistics of picture characteristics (first-order information) entailed higher-order pooling. By feature reprocessing, this method produced visual representations that were easier to identify from first-order data. The attention model was an end-to-end model [23,24] that was based on attention. This model shared similarities with the location-based algorithm in that it employed the strategy of first identifying essential components, followed by feature extraction and classification decisions. The attention model differed in that all modules were combined into a single CNN structure to acquire end-to-end training features. To address the challenges presented by the fine-grained image recognition of birds discussed earlier, a part-based identification method was proposed. This method entailed extracting features from each part of the bird and subsequently integrating those features to derive the comprehensive classification results for the bird. Furthermore, to ensure both the recognition accuracy and speed of the model, lightweight modules are incorporated to enhance its practicality.

## 2. Materials and Methods

### 2.1. Data Collection and Processing

In this study, the improved YOLOv5 model was validated using the CUB200-2011 dataset, which comprised 11,788 images, each depicting 200 species of birds. The dataset had been divided into training and testing subsets, comprising 5994 and 5794 samples, respectively. As shown in Figure 1, we tested the results of the bird recognition model through heat maps and found that the model did not pay enough attention to the heads and tails of birds.

In order to improve the bird recognition performance of the model, we used LabelImg to not only mark the left and right wings, which had high recognition accuracy, but also added the head and tail to improve the recognition ability of the model, with a total of 800 positions. Following, the PASCAL VOC format is saved in XML files, as shown in Figure 2.

Regarding detection, the performance of the part detection model might deteriorate when the sample data is limited or unbalanced. However, the CUB200-2011 dataset contains a relatively small number of images per bird, generally not exceeding 60. To bolster the model’s generalization ability, data augmentation techniques, including horizontal flipping, high brightness, and Gaussian noise, were applied to the training dataset. The effectiveness of these enhancements is illustrated in Figure 3.

### 2.2. Data Collection and Processing

#### 2.2.1. Part-Based Bird Image Recognition

This research suggested a partial detection technique in light of the difficulties posed by the substantial within-class variances and the limited between-class variations in bird images, as well as the influence of ambient factors. The enhancement of local key feature extraction from bird images was the goal of this technique. The acquired spatial data were then used to assist in the identification and categorization of these bird images. Figure 4 illustrates how bird images were identified and classified.

#### 2.2.2. YOLOv5 Algorithm

The network structure of YOLOv5 [25,26] is illustrated in Figure 5, and it is primarily divided into three parts: backbone, neck, and prediction. In YOLOv5-6.0, the backbone includes the Conv module, the C3 module, and the SPPF module.

The design of the neck network structure follows a combination of FPN and PAN structures. FPN transmits strong semantic features from top to bottom, while PAN transmits robust localization features from bottom to top. By integrating these two approaches, the model leverages the advantages of both, thereby enhancing the feature extraction capability of the model.

The prediction network structure comprises three detection layers that identify target objects of various sizes. These layers subsequently merge the positional coordinates, category information, and score information of the detected frames for the final target. The process involves detecting target objects of different sizes and then combining the characteristics of these three distinct receptive fields.

#### 2.2.3. Improved YOLOv5 Algorithm Model

The backbone network of the traditional YOLOv5 network model is primarily based on the Conv module and the C3 module for feature extraction. Among these, the C3 module is its core component. It employs one Conv with a stride of 2 and two Conv with a stride of 1 to obtain feature maps of varying sizes, enhancing the network’s receptive field and improving the model’s feature extraction capability. The neck network enhances the model’s detection performance by fusing features at different scales.

Due to the complexity and diversity of differences and similarities among bird parts, the network requires a stronger multi-scale feature extraction capability. To enhance the model’s detection performance, the Res2Net-CBAM module is introduced after each C3 module in the YOLOv5 backbone, boosting the model’s feature extraction ability. Furthermore, the CBAM module is introduced after the feature maps of each scale in the neck to enhance the model’s detection accuracy. Figure 6 and Table 1 illustrate the structural diagram of the enhanced YOLOv5 algorithm.

#### 2.2.4. Add CBAM Attention Mechanism

The CABM attention technique is introduced in this study to improve the model’s capacity to extract critical feature information. Figure 7 shows how the CABM attention mechanism [27] primarily combines the channel attention module with the spatial attention module. To produce feature maps that are more effective, the input raw feature map is subjected to the channel attention module and the spatial attention module.

Figure 8 depicts the specific structure of the channel attention module [28]. From the figure, it is evident that the input feature map undergoes global maximum pooling and global average pooling. Subsequently, it passes through an MLP (multi-layer perceptron), followed by an addition of the features output by the MLP. The resulting output then undergoes activation using the sigmoid function, resulting in the channel attention feature map. This channel attention feature map is then utilized in a multiplication process to weigh the input feature map. The outcome is a comprehensive feature map with channel attention weight. This entire process can be expressed as follows:(1)$$M_{c}\left(F\right)=\sigma \left(MLP\left(AvgPool\left(F\right)\right)+MLP\left(MaxPool\left(F\right)\right)\right)\;=\sigma \left(W_{1}\left(W_{0}\left(F_{avg}^{c}\right)\right)+W_{1}\left(W_{0}\left(F_{max}^{c}\right)\right)\right)$$

Specifically, the spatial attention module’s structural structure is shown in Figure 9. The output of the channel attention weights from the channel attention module, the whole feature map, is subjected to both global maximum pooling and global average pooling. The channel dimension is then joined by these two pooled findings. After that, the final feature map is subjected to a 7 × 7 convolutional dimensionality reduction process. Only one channel remains after the reduction of dimensions. To create the spatial attention feature map, the acquired feature map is then normalized using the sigmoid function.

Next, the spatial attention feature map is multiplied by the complete feature map with channel attention weights, resulting in a weighted spatial attention feature map. This, in turn, combines with the feature map that has both channel attention weights and spatial attention weights. The calculation formula for this process is as follows:(2)$$M_{s}=\sigma \left(f^{7\times 7}\left(\left[AvgPool\left(F\right);MaxPool\left(F\right)\right]\right)\right)\;=\sigma \left(f^{7\times 7}\left(\left[F_{avg}^{s};F_{max}^{s}\right]\right)\right)$$

#### 2.2.5. Optimize Backbone Network

The Res2Net-CBAM module, as designed in this paper, combines Res2Net [29] and CBAM to create a backbone structure, as illustrated in Figure 10. The Res2Net module establishes a hierarchical-like residual connection structure within the residual element, departing from the conventional single 3 × 3 convolution kernel. After passing through a 1 × 1 convolution, the input evenly divides the feature map into S groups based on channels. With the exception of the first group, each subsequent feature group undergoes processing by a 3 × 3 convolutional layer. Starting from the third group, the 3 × 3 convolutional layer accumulates all preceding feature information. This process results in an expanded receptive field for each output, thereby capturing multi-scale feature map information. The calculation for each output part is as follows:(3)$$Y_{i}=\left\{\begin{array}{c} {\;X}_{i},\;i=1;\\ {\;K}_{i}\left(X_{i}\right),\;i=2;\\ {\;K}_{i}\left(X_{i}+Y_{i-1}\right),\;3\leq i\leq s. \end{array}\right.$$
where $K_{i}$ represents the convolution operation, the output $Y_{i}$ of each group is concatenated, and then the output Y of the Res2Net module is obtained through 1 × 1 convolution.

To enhance the extraction ability of important feature information, the output Y of the Res2Net module is routed to the CBAM module. This is achieved after passing through both the channel attention module and the spatial attention module. Lastly, the input X of the residual unit is connected to the output of the same unit through a residual connection, resulting in the final output F of the Res2Net-CBAM module. The calculation is performed as follows:(4)$$F_{1}=M_{c}\left(Y\right)⊗Y\;F_{2}=M_{s}\left(F_{1}\right)⊗F_{1}\;F=F_{2}+X$$

Thus, the Res2Net-CBAM module proposed in this paper can effectively capture both local and global image features at a more fine-grained level. Moreover, the incorporation of residual connections helps enhance contextual information and improve the distribution and processing capacity of feature map channels, thereby increasing sensitivity to feature information.

## 3. Results

### 3.1. Evaluation Metrics

To verify the performance of the improved algorithm model, a set of metrics is introduced. Accuracy, precision, recall, and mAP are defined as follows:(5)$$Accuracy=\frac{correct\;classifications}{all\;classifications}$$
(6)$$Precision=\frac{TP}{TP+FP}$$
(7)$$Recall=\frac{TP}{TP+FN}$$
(8)$$AP=\sum_{i=1}^{n-1}\left(r_{i+1}-r_{i}\right)P_{inter}\left(r_{i}+1\right)$$
(9)$$mAP=\frac{\sum_{i=1}^{k}{AP}_{i}}{i}$$
where correct classifications are the number of correct classification results, all classifications are the total number of classification results., and accuracy is used to test bird image classification. TP represents the detection results that are true positive, meaning they are positive and actually correct. FP represents the detection results that are false positive, meaning they are negative but incorrectly identified as positive. FN stands for false negative, indicating the cases where the result is negative but should have been detected as positive. AP denotes the area enclosed by the PR curve and the coordinate axis, while *mAP* represents the mean average precision calculated across all types of AP. These measures are utilized to evaluate the detection performance of the model.

### 3.2. Experimental Setup

This article utilizes validation sets to assess training results. It gathers frame data for each segment of every image along with the corresponding bird category. Subsequently, it chooses the highest confidence value for each section of each picture to represent the distinctive information for that segment. Each image encompasses up to 4 pieces of information, with any redundant data being eliminated. The details for each bird segment obtained through the part detection network encompass part type, bird species, and confidence score. This paper distinguishes and categorizes bird images by assigning distinct weights to the four segments and aligning them with the corresponding confidence scores.

The weights of the four parts were then listed as ranges with equal differences in increments of 0.01, ranging from 0 to 1. The values within their respective ranges were combined, resulting in a total of 176,851 combinations of 1. By utilizing all of these combinations, the aforementioned test results were multiplied by the weight of each corresponding part result, factoring in its confidence level. This process yielded weighted values, enabling the identification of the site ID. Subsequently, the corresponding bird ID was obtained and used to form a dataset.

This procedure was applied to all data within each row. The summed values of the bird IDs were then compared, and if there were more than two different bird IDs, a comparison of the magnitudes of the values was conducted. The bird ID with the larger value was chosen as the final identification. The correct bird ID alignment was then verified. Consistency in bird IDs indicated correct classification, while discrepancies denoted classification errors. The classification results for each image were tallied to calculate the classification accuracy.

The aforementioned operations were performed for each weight combination, generating classification results based on different weights. These results were then compared to determine the classification accuracy. The weight combination yielding the highest accuracy represents the optimal ratio for the required part weights.

### 3.3. Experimental Setup

#### 3.3.1. Part-Based Recognition Experiment Results

The optimal weight ratio is determined through the weight allocation experiment mentioned above, which facilitates the part-based bird classification test. The system’s recognition test, as described in this chapter, was conducted on both YOLOv5 and the improved CBAM attention mechanism-based YOLOv5, utilizing the test set. Additionally, YOLOv5 was employed to train the bird dataset referred to in this paper, along with the comprehensive bird annotation file. The model’s recognition capability was tested on the same test set. The experimental results are presented in Table 2.

As depicted in Table 1, the accuracy of YOLOv5’s part-based recognition classification is 4.1% higher than that of direct recognition classification. In conclusion, the proposed part-based identification and classification method demonstrates superior performance in fine-grained bird image recognition.

#### 3.3.2. Ablation Experiment

In order to enhance the identification performance of birds, this paper proposes an improvement to the YOLOv5 model, especially for the enhancement of site recognition ability. To evaluate the effectiveness of the Res2Net-CBAM module and the CBAM module, four sets of protocols were designed to perform ablation experiments. The experimental results are shown in Table 2.

Solution 1: Based on the YOLOv5 network., the identification experiment is carried out based on the site.

Solution 2: On the basis of scheme 1, add the Res2Net-CBAM module after each C3 module in the backbone.

Solution 3: On the basis of scheme 1, add CABM modules after the last three C3 modules of the neck.

Solution 4: On the basis of scheme 1, add the Res2Net-CBAM module after each C3 module in the backbone, and add the CABM module after the last three C3 modules of the neck.

The experimental results are shown in Table 3. YOLOv5 with the Res2Net-CBAM module added to the backbone network is 1.7%, 1.9%, 2.9%, and 0.6% higher than the native YOLOv5 model in mAP@0.5, precision, recall, and accuracy, respectively. The addition of the CBAM attention mechanism module to the neck is 0.8%, 0.7%, 1.9%, and 0.3% higher than the original network in mAP@0.5, precision, recall, and accuracy, respectively. The combined model is 2.3%, 2.2%, 4.9%, and 1.2% higher than the improved model in mAP@0.5, precision, recall, and accuracy, respectively.

#### 3.3.3. Model Comparison

To assess the effectiveness of the proposed improved method, we conducted comparative experiments with strongly supervised algorithms and weakly supervised algorithms (e.g., PN-CNN, SPDA-CNN, FCAN, FT, etc.). Table 4 below presents the experimental results of the proposed bird image recognition method on the CUB200-2011 dataset.

The table illustrates that the recognition accuracy of the proposed method for the CUB200-2011 dataset is 1.2%, 1.5%, 1.9%, 3.6%, and 3.8% higher, respectively, than that of the strongly supervised algorithm on the same dataset. Additionally, it is 2.5%, 6.4%, 1.6%, and 1.0% higher than the weakly supervised algorithms FT, PC, AM-CNN, and MS-SRP-D, respectively.

## 4. Discussion

Aiming for fine-grained recognition of bird images, this paper proposes a part-based recognition method that locates four regions of the bird: the head, left wing, right wing, and tail. This approach aims to fully extract key regional features of the object. Building upon the YOLOv5 network, the site detection network has been improved to ensure high accuracy while maintaining recognition speed.

By incorporating the Res2Net-CBAM module into the backbone network, the proposed model enhances the channel’s self-regulation ability and improves the extraction of key feature information with reduced computation. This adjustment results in a larger receptive field at scale. Additionally, the CBAM module has been added to the neck to augment the extraction of key information across low, medium, and high layers of network features. These enhancements collectively improve the detection performance of bird parts and enhance the model’s convergence ability. 

In order to further understand the performance of the model, we drew comparison charts of precision, recall, and mAP@0.5 curves based on the experimental results in Figure 11. It can be concluded from the figure that the precision curve of the model proposed in this paper not only rose quickly but also had strong stability, indicating that the model has stronger anti-noise ability and stronger robustness. Through the recall curve chart, we can find that our model had a fast-rising speed and a high peak value, which proved that the improved model has a stronger ability to extract key features. According to the mAP@0.5 curve, we can see that the improved model had the strongest upward trend and speed, which proved that the model has faster convergence speed and stronger detection performance.

In order to visually represent the model’s feature extraction ability for fine-grained images, Grad-Cam [35] was utilized to visualize the features. It can be observed from Figure 12 that due to the presence of the annotation box, the model can effectively determine the location of key information. However, environmental factors still influence the model’s extraction of key features. The enhanced model improves attention towards key features and mitigates the impact of the environment.

Simultaneously, the experimental results show that compared to the original YOLOv5 model, the proposed model exhibits a 2.2% increase in mAP@0.5, a 1.2% increase in precision, and a 6.3% increase in recall. Furthermore, the accuracy of bird identification is enhanced by 1.2%, thereby demonstrating the performance of the improved model and the effectiveness of the proposed method.

Analysis of other fine-grained bird image algorithms shows that SPDA-CNN utilizes more part-labeling information to enhance the performance of the detection network, thereby improving the network’s classification ability. However, the model employed in this paper utilizes less part-labeling information to achieve a more effective detection effect and classification ability. FCAN captures essential feature information by employing the attention region. This paper further enhances the network by incorporating an attention module based on annotated information, resulting in more efficient extraction of key feature information. MS-SRP-D employs the SRP module to enhance the model’s capability to extract crucial feature information from images. In contrast, this paper demonstrates a stronger ability to extract key feature information by initially detecting parts and enhancing the part-detection network.

## 5. Conclusions

In this paper, we propose a method that involves extracting features from each part of the bird and then merging these partial features to classify the bird image. This approach addresses the challenges posed by large within-class differences and smaller between-class differences among bird species. Considering that this method requires a high amount of computation for feature extraction on-part, we further enhance the network and obtain a fine-grained image classification model of birds with fast recognition speed and high accuracy. In future research, we will consider using ant colony optimization [36,37] to improve the model’s ability to extract key features of parts. Moreover, as spatial position information of each part of the bird is acquired during the feature extraction process, future research could also explore comparing this positional information with the bird’s body to infer its action posture.

## Figures and Tables

**Figure 1 sensors-23-08204-f001:**
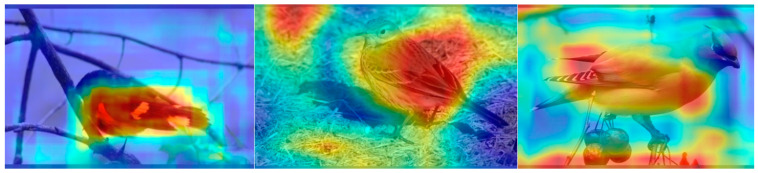
Feature Visualization of Bird Images.

**Figure 2 sensors-23-08204-f002:**
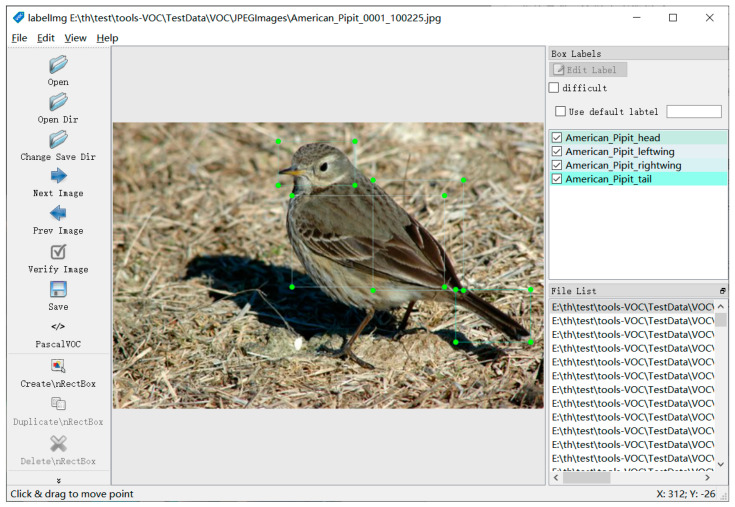
Bird’s four parts(head, leftwing, rightwing, tail) labeling.

**Figure 3 sensors-23-08204-f003:**
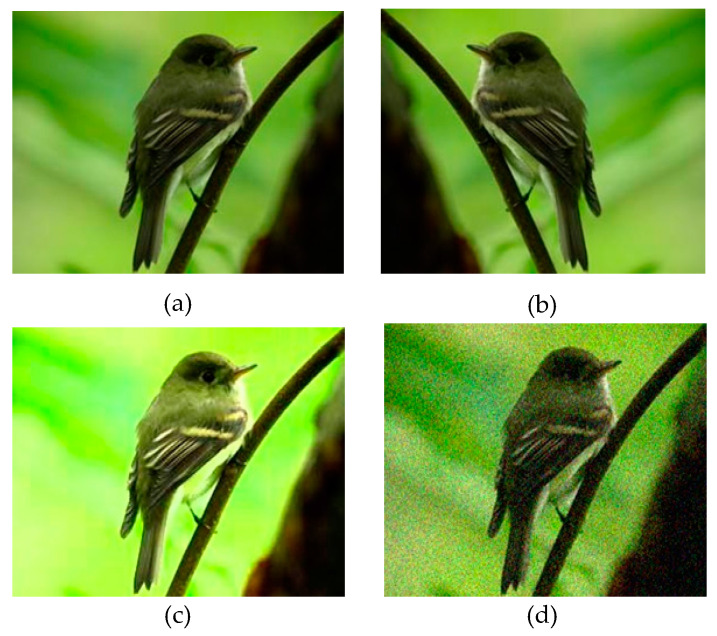
Data Augmentation. (**a**) Original. (**b**) Horizontal flip. (**c**) High brightness. (**d**) Gaussian noise.

**Figure 4 sensors-23-08204-f004:**
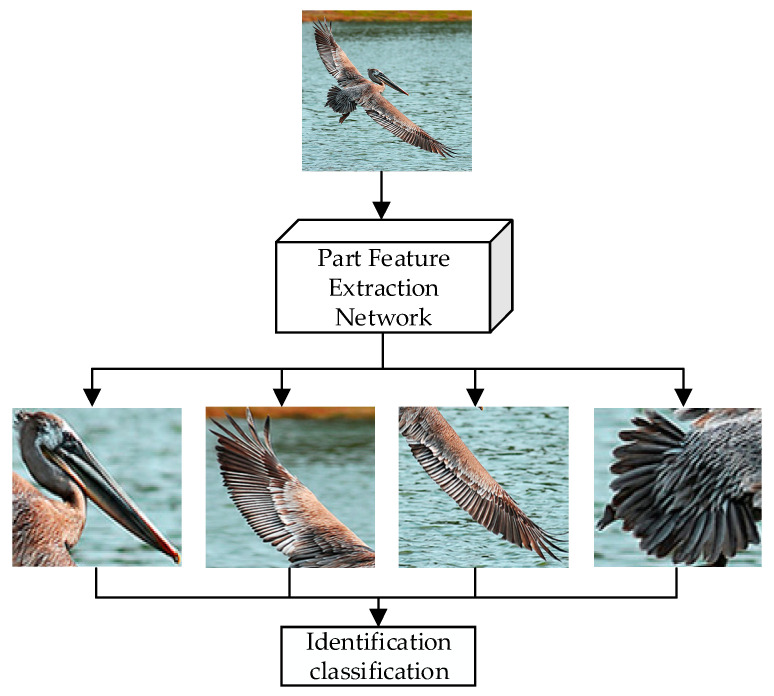
Part-Based Bird Image Recognition.

**Figure 5 sensors-23-08204-f005:**
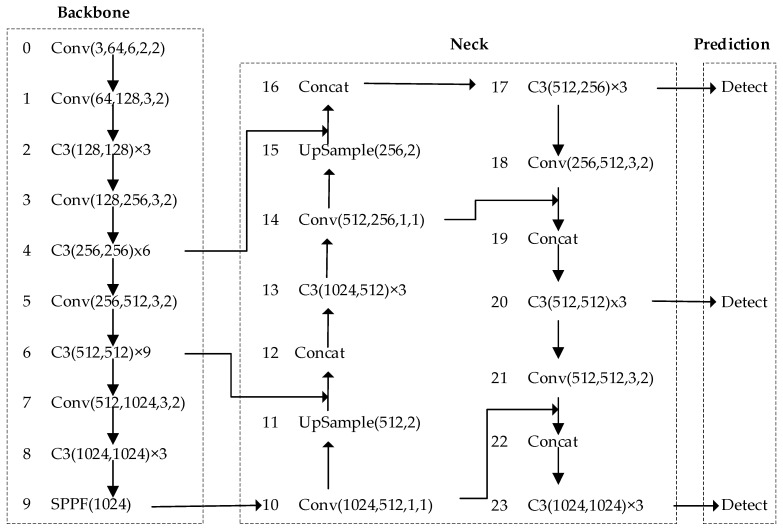
YOLOv5-6.0 Algorithm Structure.

**Figure 6 sensors-23-08204-f006:**
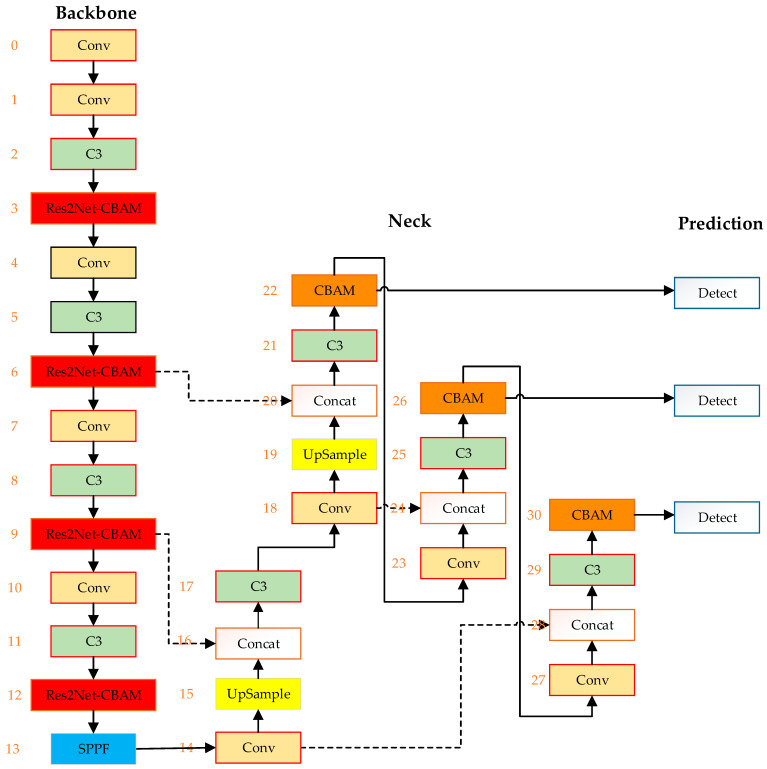
Improved YOLOv5 Algorithm Structure.

**Figure 7 sensors-23-08204-f007:**
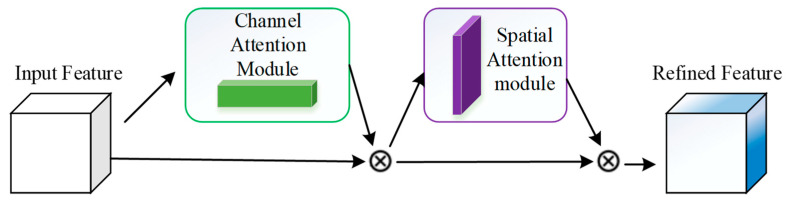
CBAM Attention Module.

**Figure 8 sensors-23-08204-f008:**
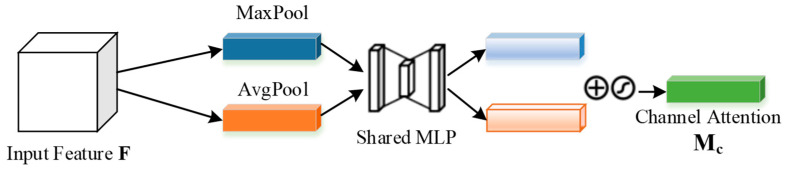
Channel Attention Module.

**Figure 9 sensors-23-08204-f009:**
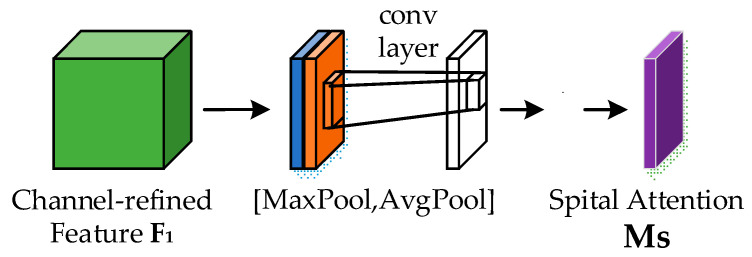
Spatial Attention Module.

**Figure 10 sensors-23-08204-f010:**
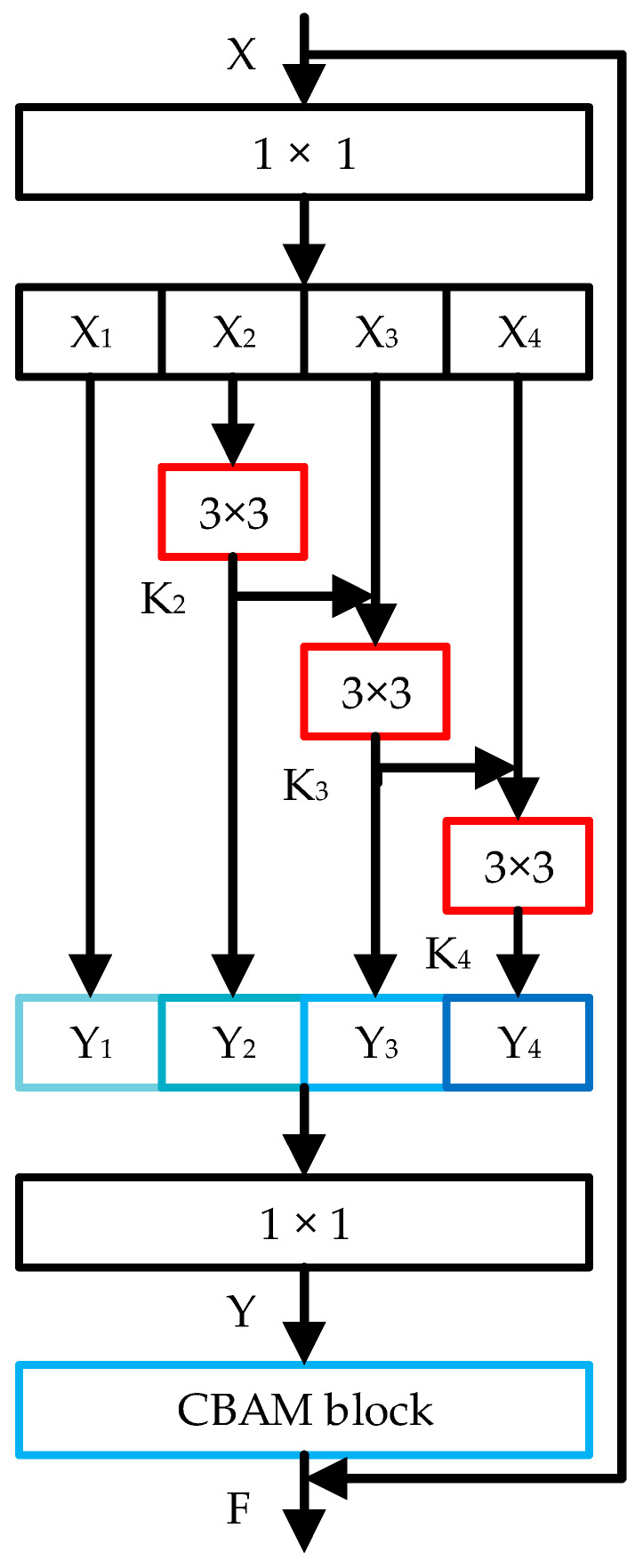
Res2Net-CBAM Module.

**Figure 11 sensors-23-08204-f011:**
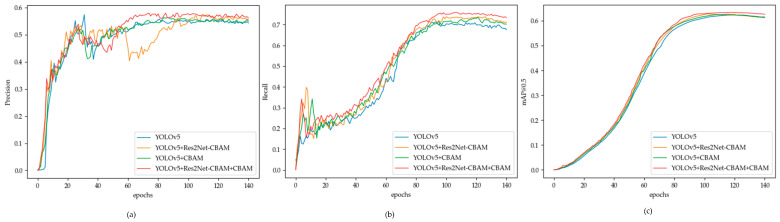
Experimental results curve charts: (**a**) Precision; (**b**) Recall; (**c**) mAP@0.5.

**Figure 12 sensors-23-08204-f012:**
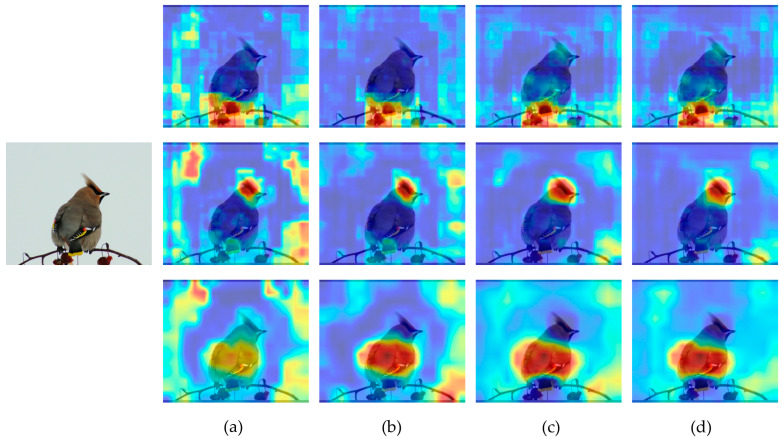
Feature Visualization for Different Detection Layers. (**a**) Original YOLOv5. (**b**) YOLOv5 + CBAM. (**c**) YOLOv5 + Res2Net-CBAM. (**d**) YOLOv5 + Res2Net-CBAM + CBAM.

**Table 1 sensors-23-08204-t001:** Improved YOLOV5 network structure.

Layer	Module	From	Number	Arguments	Input	Output
0	Conv	−1	1	(3, 80, 6, 2, 2)	640 × 640 × 3	320 × 320 × 80
1	Conv	−1	1	(80, 160, 3, 2)	320 × 320 × 80	160 × 160 × 160
2	C3	−1	4	(160, 160)	160 × 160 × 160	160 × 160 × 160
3	Res2Net-CBAM	−1	1	(160, 160)	160 × 160 × 160	160 × 160 × 160
4	Conv	−1	1	(160, 320, 3, 2)	160 × 160 × 160	80 × 80 × 320
5	C3	−1	8	(320, 320)	80 × 80 × 320	80 × 80 × 320
6	Res2Net-CBAM	−1	1	(320, 320)	80 × 80 × 320	80 × 80 × 320
7	Conv	−1	1	(320, 640)	80 × 80 × 320	40 × 40 × 640
8	C3	−1	12	(640, 640)	40 × 40 × 640	40 × 40 × 640
9	Res2Net-CBAM	−1	1	(640, 640)	40 × 40 × 640	40 × 40 × 640
10	Conv	−1	1	(640, 1280, 3, 2)	40 × 40 × 640	20 × 20 × 1280
11	C3	−1	4	(1280, 1280)	20 × 20 × 1280	20 × 20 × 1280
12	Res2Net-CBAM	−1	1	(1280, 1280)	20 × 20 × 1280	20 × 20 × 1280
13	SPPF	−1	1	(1280, 1280, 5)	20 × 20 × 1280	20 × 20 × 1280
14	Conv	−1	1	(1280, 640, 1, 1)	20 × 20 × 1280	20 × 20 × 640
15	UpSample	−1	1	(2, ‘nearest’)	20 × 20 × 640	40 × 40 × 640
16	Concat	(−1, 9)	1	(1)	40 × 40 × 640	40 × 40 × 1280
17	C3	−1	4	(1280, 640, False)	40 × 40 × 1280	40 × 40 × 640
18	Conv	−1	1	(640, 320, 1, 1)	40 × 40 × 640	40 × 40 × 320
19	UpSample	−1	1	(2, ‘nearest’)	40 × 40 × 320	80 × 80 × 320
20	Concat	(−1, 6]	1	(1)	80 × 80 × 320	80 × 80 × 640
21	C3	−1	4	(640, 320, False)	80 × 80 × 640	80 × 80 × 320
22	CBAM	−1	1	(320)	80 × 80 × 320	80 × 80 × 320
23	Conv	−1	1	(320, 320, 3, 2)	80 × 80 × 320	40 × 40 × 320
24	Concat	(−1, 18)	1	(1)	40 × 40 × 320	40 × 40 × 640
25	C3	−1	4	(320, 320, False)	40 × 40 × 640	40 × 40 × 640
26	CBAM	−1	1	(640)	40 × 40 × 640	40 × 40 × 640
27	Conv	−1	1	(640, 640, 3, 2)	40 × 40 × 640	20 × 20 × 640
28	Concat	(−1, 14)	1	(1)	20 × 20 × 640	20 × 20 × 1280
29	C3	−1	4	(1280, 1280, False)	20 × 20 × 1280	20 × 20 × 1280
30	CBAM	−1	1	(1280)	20 × 20 × 1280	20 × 20 × 1280
31	Detect	(22, 26, 30)	1	(800, ((10, 13, 16, 30, 33, 23), (30, 61, 62, 45, 59, 119), (116, 90, 156, 198, 373, 326)))	80 × 80 × 320 40 × 40 × 640 20 × 20 × 1280	80 × 80 × 3 × 805 40 × 40 × 3 × 805 20 × 20 × 3 × 805

**Table 2 sensors-23-08204-t002:** Performance Comparison Experiment of Bird Image Recognition.

Methods	Part-Based YOLOv5	Whole Based YOLOv5
Accuracy (%)	85.4	81.3
Time per inference (ms)	25.9	25.4

**Table 3 sensors-23-08204-t003:** Ablation Experiment Results.

Solution	Accuracy (%)	mAP@0.5 (%)	Precision (%)	Recall (%)
YOLOv5	85.4	61.0	55.5	70.7
YOLOv5 + Res2Net-CBAM	86.0	62.7	57.4	73.8
YOLOv5 + CBAM	85.7	61.8	56.2	72.8
YOLOv5 + Res2Net-CBAM + CBAM	86.6	63.3	57.7	75.6

**Table 4 sensors-23-08204-t004:** Bird Recognition Accuracy Comparison.

Method	Accuracy (%)	Label
Ours	86.6	BBox + Parts
PN-CNN [8]	85.4	BBox + Parts
SPDA-CNN [17]	85.1	BBox + Parts
FCAN [30]	84.7	BBox
MG-CNN [31]	83.0	BBox
PA-CNN [9]	82.8	BBox
FT [32]	84.1	No
PC [33]	80.2	No
AM-CNN [10]	85.0	No
MS-SRP-D [34]	85.6	No

## Data Availability

We used CUB-200-2011 birds datasets.

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
