# Peer review of "Research on Fine-Grained Image Recognition of Birds Based on Improved YOLOv5"

_sensors, 2023, doi:10.3390/s23198204_

Round 1

Reviewer 1 Report

The main proposed approach has novelty in contribution and methodology. Revision in terms of technical details is needed before publication. Also, paper organization can be improved. In this respect, some comments are suggested to describe technical details.

1. The main aim of this paper is bird recognition. What is the benefit of this research in real applications? Where it can be used? Discus briefly.

2. The used deep network has 3 parts. But, what is the relation of these parts with bird parts in the real image (Figure 1)? In the other words how do you segment the bird image to different parts? Discuss with technical details

3. As can be seen in the figure 4, three different detect node is used in the prediction module. How do you make the final decision? Do you use majority? How do you make decision when all three nodes suggest a different label?

4. The current proposed method can be used widely in computer vision applications such as medical diagnosis or image retrieval. For example, I find a paper titled “Innovative local texture descriptor in joint of human-based color features for content-based image retrieval”, which has enough relation. Cite this paper and some other papers as possible future work ideas.  

5. Describe the input and output size of all modules in the figure 4 (Table format is suggested).   

6. Describe the hyper-parameter optimization process of the used YOLO network.

Author Response

  1. Added introduction to practical applications. On page1, line 34,35,36
  2. Added reasons why four parts were chosen. On page1, line96, page2, line 97,98
  3. Determine the final classification result by adding weight to each part and the confidence of the detected part itself. In Section 3.2
  4. Added future ideas for ant colony optimization. In Section 5
  5. Added input and output details of each module of the model. On page 7
  6. Only batch-size has been changed for the hyperparameters, and other parameters are the default.

Reviewer 2 Report

The authors have proposed a detection framework based on YOLOv5 for bird recognition, which includes enhancements to the YOLOv5 backbone network through the integration of an attention-based module. Below are my comments:

1) Extensive editing of the English language is required, including but not limited to:

(a) On page 2, line 72, the phrase "from beginning to end" appears awkward. It may be implied that RNNs can handle sequential features, but the expression needs refinement.

(b) On page 3, line 96, the term "the damage recognition model" is unclear. It requires clarification.

(c) On page 3, line 100, the word "highlighting" is unusual in this context. Did you mean "high-contrast"?

2) I have some general questions about the manuscript. Firstly, I'd like to inquire about the term "classification." It is traditionally discussed separately from "detection." (a) Is it implemented separately or within the YOLOv5 framework? (b) Could you clarify the classes used in your experiments? I assume they correspond to the categories mentioned on page 3, line 94, but it's not explicitly stated.

3) The explanation of your proposed model lacks clarity. This pertains not only to the YOLOv5 structure but also to the overall workflow, including inputs and corresponding outputs. Even though Figure 2 is provided, it doesn't sufficiently explain the significance of the four sub-images or how the identification classification process functions (which is also an awkward term).

4) Regarding the dataset you used, it appears you created new labeling information using the LabelImg tool. I strongly recommend including additional figures that illustrate what you created. This would offer valuable insights into the inputs and outputs of the dataset you utilized.

5) The experimental section requires significant improvement. One critical issue is the limited number of figures provided in this section. Detection results, particularly qualitative results, are vital evidence to support your arguments and statements. Building on my previous comments, there are doubts about the implementation of your proposed model, not to mention the results of various detection frameworks presented in Table 3. Including more results in various formats would enhance readers' understanding and make the article more convincing.

Comments related to the quality of English language has been provided above.

Author Response

  • (a)Modified to ‘end-to-end’ on page1, line 73,74 (b)Modified to ‘part detection model’ on page 3, line 108 (c) Modified to ‘high brightness’ on page 3, line 112, page4 Figure 3
  • (a) The parts are detected through YOLOv5, and then classification is achieved by weighted calculation of the confidence of each detected part. (b) In Section 2.1, four parts of each bird are explained, with a total of 800 categories that need to be detected.
  • Section 2.2.3 adds a detailed table of each module of the model (page 7 ), and adds the reasons why these four parts were selected in section 2.1. We implement weighted calculations on these four parts to achieve the identification and classification of birds. The details are in 3.2 Section
  • Section 2.1 adds content on how we choose parts on page 3, line 97-105
  • Chapter 4 adds Precision, Recall, [email protected] curve comparison chart and related discussion instructions on page 13, line 341-350

Round 2

Reviewer 1 Report

The authors' answers to the comments are satisfactory. A more complete explanation has been added to the text, which explains the presented method more clearly. 

Reviewer 2 Report

No further comments.

No further comments.